# Centromere evolution and CpG methylation during vertebrate speciation

Kazuki Ichikawa[1], Shingo Tomioka[1], Yuta Suzuki [ID] [1], Ryohei Nakamura[2], Koichiro Doi[1], Jun Yoshimura[1], Masahiko Kumagai [ID] [2], Yusuke Inoue[2], Yui Uchida[2], Naoki Irie [ID] [2], Hiroyuki Takeda[2] & Shinich Morishita[1]

Centromeres and large-scale structural variants evolve and contribute to genome diversity during vertebrate speciation. Here, we perform de novo long-read genome assembly of three inbred medaka strains that are derived from geographically isolated subpopulations and undergo speciation. Using single-molecule real-time (SMRT) sequencing, we obtain three chromosome-mapped genomes of length ~734, ~678, and ~744Mbp with a resource of twenty-two centromeric regions of length 20–345kbp. Centromeres are positionally conserved among the three strains and even between four pairs of chromosomes that were duplicated by the teleost-specific whole-genome duplication 320–350 million years ago. The centromeres do not all evolve at a similar pace; rather, centromeric monomers in non-acrocentric chromosomes evolve significantly faster than those in acrocentric chromosomes. Using methylation sensitive SMRT reads, we uncover centromeres are mostly hypermethylated but have hypomethylated sub-regions that acquire unique sequence compositions independently. These findings reveal the potential of non-acrocentric centromere evolution to contribute to speciation.

[1] Department of Computational Biology and Medical Sciences, Graduate School of Frontier Sciences, The University of Tokyo, 5-1-5 Kashiwanoha, Kashiwa, Chiba 277-8583, Japan. [2] Department of Biological Sciences, Graduate School of Science, The University of Tokyo, 7-3-1 Hongo, Bunkyo-ku, Tokyo 113-0033, Japan. Kazuki Ichikawa, Shingo Tomioka, Yuta Suzuki, Ryohei Nakamura and Koichiro Doi contributed equally to this work. Correspondence and requests for materials should be addressed to H.T. (email: htakeda@bs.s.u-tokyo.ac.jp) or to S.M. (email: moris@edu.k.u-tokyo.ac.jp)

Revision of the draft genomes that contain many gaps has been attracting tremendous interest in recent years given that long reads (>10 kbp) may now be obtained using single-molecule real-time (SMRT) sequencing[1–3]. Several software programs, including Celera[4], PBcR[5], HGAP[6], FALCON[7], DALIGN[8], and MHAP[9] have been used to sequence the bacterial[6], human[7,9,10], gorilla[11], *Oropetium thomaeum*[12], and seabass[13] genomes. Each assembled genome features extremely long contigs and has fewer number of gaps than do the Sanger sequences of genomes of the same species. In particular, typical bacterial genomes have no gaps[6]. Highly accurate long contigs have been useful in enumeration of structural variants (SVs)[7,9–11,14], filling gaps such as centromeres[12], extending contigs to telomeres[12], and phasing haplotypes[15]. The medaka, Japanese killifish (*Oryzias latipes*), has been the subsect of research for nearly a century. This research has achieved the first demonstration of X and Y chromosome exchange through crossover[16], and provided many insights into developmental biology, reproduction biology, and genome science, owing partly to many useful biological characters that medaka shares with zebrafish[17,18]. Using Sanger sequencing, we reported the version 1 of the medaka reference genome, which has an estimated size of ~800 Mb, from the Hd-rR inbred strain in 2007[19]; however, this version contained low-quality regions and 97,933 sequence gaps. To overcome these deficiencies, we collected long SMRT reads from the three inbred strains, thereby generating extremely long assembled contigs.

In long-read genome assemblies, it is feasible to list centromeres and large-scale SVs and consider how they evolved. To this end, we re-sequenced the genomes of three medaka inbred strains derived from different local subpopulations and listed SVs; HNI from northern Japan, Hd-rR from southern Japan and HSOK from east Korea (Fig. 1a). The estimated date of divergence for the two Japanese strains, Hd-rR and HNI, is ~18 million years ago (MYA)[20], and that for the Japanese and Korean strains is ~25 (MYA)[20]. These lineages are separated by an appropriate evolutionary distance that is close enough to reliably align noncoding sequences, but also entails sufficient sequence variations[17,19,21–23]. These subpopulations were originally considered as a single species, *Oryzias latipes*, as they can mate and produce healthy offspring under laboratory conditions. However, over a long period of geographical separation, they have

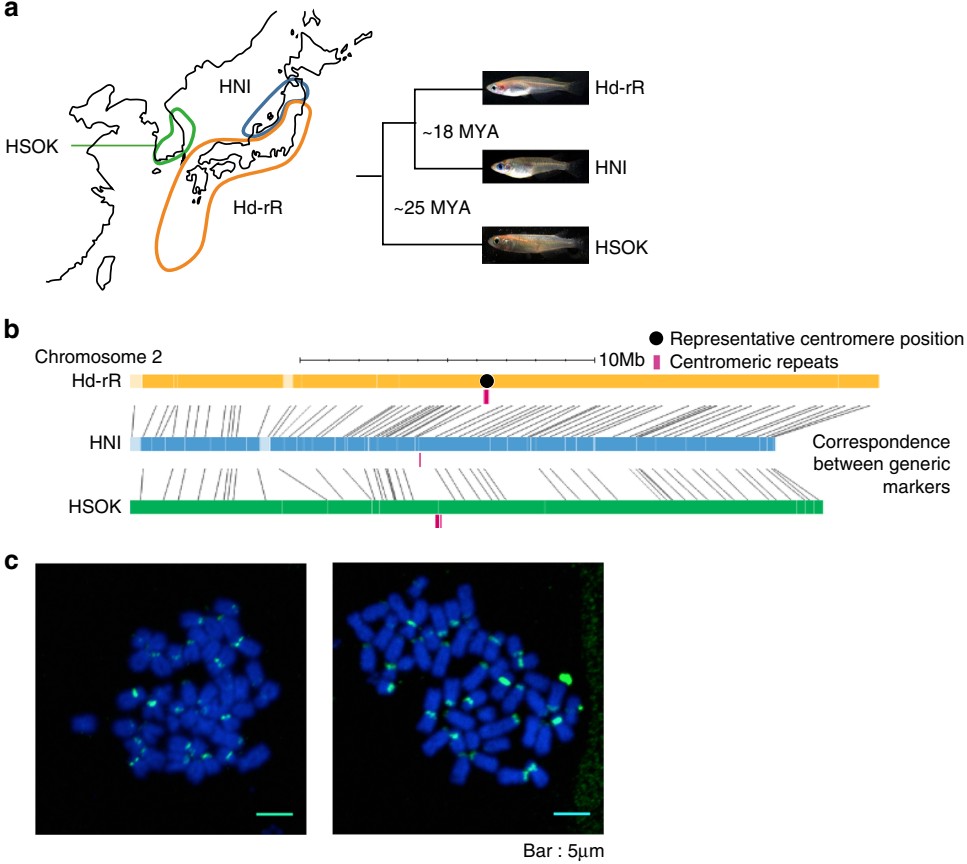

**Fig. 1** Genome assembly and analysis of centromeric repeats. **a** Hd-rR, HNI, and HSOK are inbred strains derived from southern and northern populations of Japan, and east Korea population. **b** The orange boxes are scaffolds for Hd-rR, and the other blue and green boxes are contigs of HNI and HSOK. Red bars below contigs display centromeric repeats identified (see the details in Supplementary Table 12). The gray lines connecting contig boxes reflect the correspondence between genetic markers anchored on contigs. In ten of twelve chromosomes in which centromeric repeats were sequenced, centromeric repeats were located at identical genetic loci (denoted by black solid circles) between multiple strains (Supplementary Table 12 and Supplementary Fig. 2), and the figure shows chromosomes 2. Most of contigs are oriented, but light-colored boxes remain non-oriented in chromosomes because they have only single markers, or sets of genetic markers at the same genetic distance. **c** Fluorescence in situ hybridization (FISH) images of metaphase. Probes were stained green and DNA blue with DAPI. We took two images independently and observed hybridization signals in all chromosomes except for a couple of chromosomes. We designed four different centromere-specific DNA probes from centromeric repeats (Methods), and Supplementary Fig. 3 shows FISH images for individual four DNA probes

accumulated genetic mutations, leading to phenotypic diversity. The three subpopulations are now thought to be in the middle of speciation, and indeed, the two Japanese subpopulations were recently proposed to be distinct species[24]. As such, inbred strains established from these local populations provide a unique platform for analyzing the genomic basis of vertebrate speciation.

In the present study, we report the long-range higher-order structure of medaka centromeres and correlation of SVs with differential gene expression during medaka speciation. Centromeres are positionally conserved among the three strains. The centromeres do not all evolve at a similar pace. Non-acrocentric monomers evolve more rapidly than acrocentric monomers, yielding hypomethylated regions with distinct sequence compositions. A number of insertions upstream of transcription start sites increase the GC ratio and CpG ratios, and lower DNA methylation levels, leading to significantly elevated transcriptional expression or de novo transcription.

## Results

**Generating long contigs using SMRT sequencing.** We collected DNA from adult medaka testes of the Hd-rR, HNI, and HSOK strains. We used a SMRT sequencer (PacBio RS II) to collect ~13.4, ~14.8, and ~5.5 million subreads, with average lengths of 6,519 bp, 3,575 bp, and 10,972 bp, from the Hd-rR, HNI, and HSOK strains, respectively (Supplementary Table 1). The three data sets are equivalent to coverages of ~109-, ~66.0-, and ~75.8-fold, assuming a medaka genome size of 800 Mbp. We used the FALCON assembler[7] to generate contigs (Supplementary Fig. 1); the respective N50 contig lengths were ~2.5, ~1.3, and ~3.5 Mbp (Supplementary Table 2). We polished the assembled contigs using Quiver[6]. We next used Illumina-derived short reads to correct any remaining errors; we employed Pilon[25] to this end (see Methods). Next, we compared the new Hd-rR assembly with the medaka genome version 1 that we had earlier generated by using Sanger sequencing technology[19], and confirmed the high-level sequence identity (99.8%). To assess the large-scale orderings of regions in the contigs, we explored whether the 19,448 pairs of BAC-end Sanger reads mapped approximately to the identical Hd-rR contigs in order. Only 0.3% of BAC-end pairs were inconsistent, confirming that the assembled contigs were of high quality. We also evaluated genome quality using CEGMA by checking whether a set of 458 highly conserved eukaryotic genes mapped to the contigs. We observed 87.1%, 86.3%, and 88.3% complete and 99.6%, 99.2%, and 99.6% partial matches of the genes in the Hd-rR, HNI, and HSOK genomes, respectively (Supplementary Table 3).

**Chromosome map construction.** We used 2,347 single-nucleotide polymorphism (SNP) genetic markers to construct a chromosomal map of the Hd-rR strain[19]. Assuming that genetic markers are distributed uniformly, a marker would be available every ~341kbp. Some 90% of contigs were sufficiently long to bear genetic markers; the respective N90 contig lengths of Hd-rR, HNI and HSOK were ~653, ~450, and ~1,102kbp (Supplementary Table 4). Thus, we skipped the traditional step of connecting contigs into longer scaffolds, instead attempting to directly anchor contigs to the 24 medaka chromosomes using genetic markers (Fig. 1a; Methods).

Certain contigs failed to be anchored to any chromosomes because they did not contain genetic markers. For Hd-rR contig anchoring, we used 48,955 BAC-end pairs and 199,657 fosmid-end pairs that had earlier been collected[19]. By scaffolding Hd-rR contigs connected by multiple BAC/fosmid-end pairs, we were able to anchor additional 23 Hd-rR contigs to chromosomes (Methods). A total of 768 BAC-end pairs and 376 fosmid-end

pairs linked the Hd-rR contigs. This suggests that the gaps between contigs are likely to be longer than fosmid clones of median length 37.5kbp, and longer reads would be needed to fill such gaps (Supplementary Notes). We used Hi-C data to locate 11 orphan contigs which could not be anchored onto chromosomes (Methods). We finalized the draft genomes by inserting a 1kbp gaps between neighboring contigs; we term these drafts version 2.2.4. In this version, the total numbers of bases in the contigs anchored to the Hd-rR, HNI, and HSOK chromosomes were ~733.5, ~677, and ~744 Mbp respectively with 491, 717 and 318 gaps (Supplementary Tables 4 and 5). Thus, the quantity of gaps was dramatically lower than the ~100,000 gaps in the previous Sanger-sequence Hd-rR genome assembly.

To demonstrate the comprehensive nature of our current sequences, we examined the distributions of *Tol*2 element insertions. *Tol*2 is 4682 bp in length, and represents an example of an early innate autonomous transposon in a vertebrate genome[26]. While the previous Sanger-sequence genome assembly had no full *Tol*2 matches, the new Hd-rR, HNI and HSOK genomes bore 15, 5, and 16 full matches, respectively, in different positions. These occurrences were >99.4% identical to the reference *Tol*2 sequence (Supplementary Table 6), implying their horizontal transfer after the divergence of Hd-rR and HNI (Supplementary Notes). Another example is the Y-specific region carrying *DMY*, the male-determining gene, the first non-mammalian equivalent of *SRY*[27]. *DMY* had mapped to three scaffolds with gaps in the earlier Hd-rR genome (version 1) because of its proximal repetitive elements[28], but we obtained a single contig bearing *DMY* in the version 2.2.4 (Supplementary Table 7).

**Genetic divergence and gene annotations.** In 2007, we reported that the SNP rate between the Hd-rR and HNI genomes was 3.42%[19]. However, this was an overestimate because of the low quality of the prior HNI genome assembly, and we revised the SNP rate to 2.455% based on analysis of the new genomes. Similarly, we revised the previous indel rate, of 0.594%, down to 0.424% (Supplementary Table 8). We newly generated a medaka gene model using 100 bp paired-end strand-specific RNA-Seq data that were collected from 57 developmental stages of the d-rR strain, an ancestor of Hd-rR, using Illumina Hiseq4000 (Supplementary Table 9). We assembled the RNA-Seq data in each sample using Trinity[29], and carried out gene annotation using the MAKER2[30] pipeline to predict 29,267 genes with 5' and/or 3' UTRs (Methods; Supplementary Table 10).

**Centromere evolution.** Analysis of centromeres in vertebrate genomes has been challenging[12,31–38]. Recently, three centromeres were sequenced in the ~245 Mb *Oropetium thomaeum* genome using long SMRT reads[12]. However, centromeres remain rarely sequenced in vertebrate genomes. The longest sequences of centromeric higher-order repeats in the present reference human genome is no more than 40-kbp in size[34,35]. Characterizing centromeric repeats and their higher-order repeats from Sanger reads and long reads has been only partially successful[37,38]. Once speciation is completed, representative centromeric monomers are highly diversified among 282 species[36]; however, centromere evolution during speciation and its relevance with speciation are unknown.

We first quantified centromeric satellites by searching raw PacBio subreads for a representative medaka centromeric monomer (Methods). The genomic fraction of centromeric monomers in the HSOK genome is ~2% while that in the other two genomes is ~1% (Supplementary Table 11), which accords with that centromeric array on a single chromosome varies in size

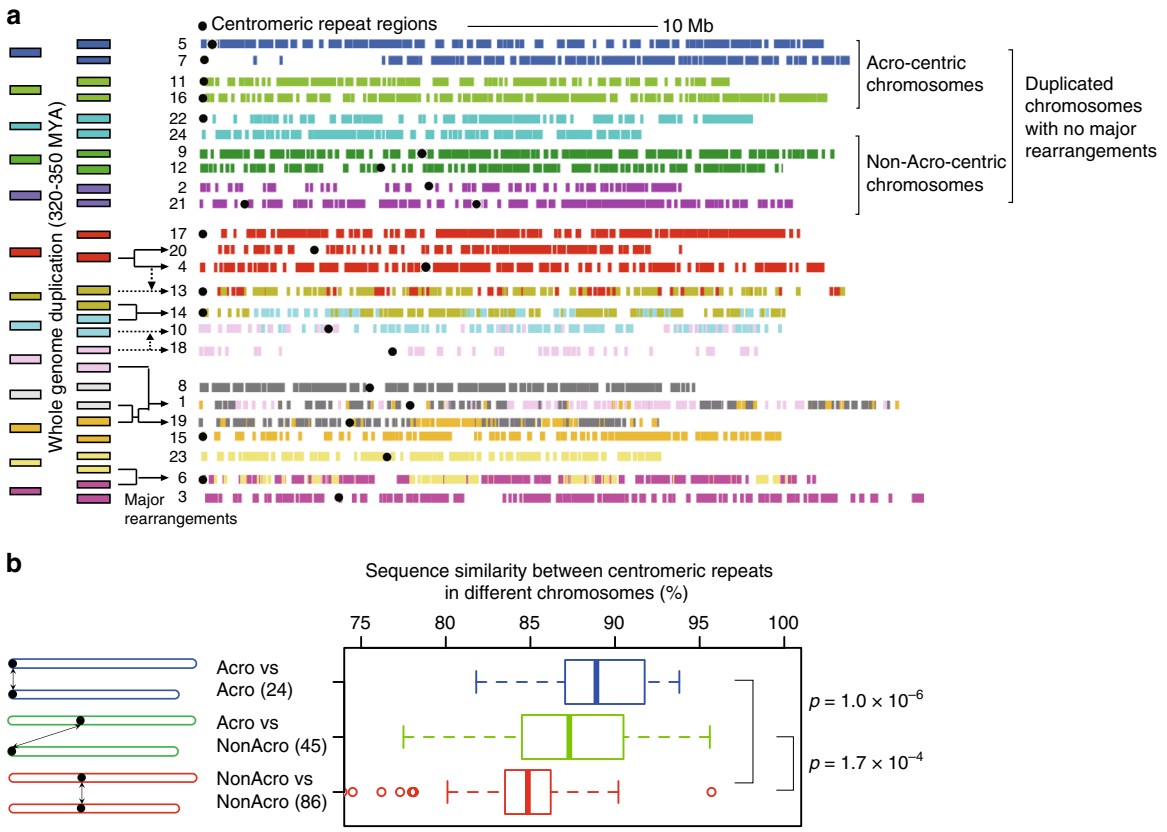

**Fig. 2** Centromere evolution. **a** Chromosome evolution from the teleost ancestral chromosomes to the medaka chromosomes. Thirteen proto-chromosomes were duplicated by the teleost WGD, and individual ancestral chromosomes are colored differently. Afterwards, major rearrangements (fissions and fusions: solid arrows, translocations: dotted arrows) shaped the medaka genome. Colored bars denote ancestral chromosome segment before the teleost WGD (see details in Supplementary Fig. 2). Representative centromere positions are depicted by black circles. Chromosome 21 has two centromeric satellite arrays at acrocentric and metacentric positions represented by 21a and 21 m respectively. We failed to find centromeric repeats in chromosome 24. For better readability, ten chromosomes (7, 11, 22, 9, 12, 13, 18, 8, 1, and 19) are inverted so that the centromere of each chromosome is in the left part. **b** The left portion shows a schematic model for illustrating three patterns of centromeric repeat comparisons between different chromosomes. "Acro vs Acro" in the top denotes comparisons between acrocentric chromosomes, "Acro vs nonAcro" between acrocentric and non-acrocentric ones, and "nonAcro vs nonAcro" between non-acrocentric ones. Pairs of all monomer clusters and their best matching clusters are grouped into three patterns. The box plots show the distributions of sequence similarities of best matching pairs in the three patterns. The similarity distribution between non-acrocentric monomer clusters "nonAcro vs nonAcro" are significantly lower than the other two patterns ($p = 1.0 \times 10^{-6}$ and $p = 1.7 \times 10^{-4}$ by Wilcoxon's rank sum test)

up to twenty-fold among individuals[37]. Searching the genomes for the representative monomers, we captured centromeric monomer sequences (Fig. 1b; Supplementary Fig. 2; Supplementary Tables 12 and 13). To validate these monomers derived from the centromeres, we designed centromere-specific DNA probes, performed fluorescence in situ hybridization (FISH) experiments (Methods), and observed signals at the centromeres of ~22 chromosome pairs that were largely consistent with their positions in sequenced genomes (Fig. 1c, Supplementary Fig. 3d). We obtained an unprecedented resource of centromeric repeats of length 20–345 kbp in vertebrates.

We analyzed centromere evolution during speciation by comparing syntenic chromosomes between the strains. We found that after the divergence ~25 MYA, centromeres were nearly positionally conserved at chromosomal syntenic regions, which allowed us to determine the representative position to each chromosome (Fig. 1b and Supplementary Fig. 2). Positional conservation of centromeres motivated us to examine their evolution after the teleost-specific whole-genome duplication (WGD) event 320–350 MYA (Fig. 2a)[39]. We previously showed that five pairs of duplicated chromosomes underwent no major rearrangements after the WGD event[19], and investigated four

pairs with centromeric repeats in detail (no centromeric regions were sequenced for one chromosome in one pair). Intriguingly, all the four pairs had nearly conserved centromere positions (Fig. 2a). In contrast, the centromere positions in the other chromosomes may have been shuffled by major rearrangements after the WGD. Thus, the position of centromeres tends to be preserved unless chromosomal arrangement took place on a large scale, and indeed it was maintained for 320–350 MY in intact chromosomes of the teleost lineage.

Positional conservation of centromeres, however, does not imply sequence conservation of centromeres. Melters et al. selected a single representative centromeric monomer from each of 282 species, and observed that the average sequence similarity between those representative centromeric monomers dropped rapidly down to 25% if the species diverged >50 MYA[36]. This suggests a possible role of centromere sequence evolution in speciation via reproductive barrier at meiotic chromosomal pairing. Since we obtained long centromeric repeats located in medaka chromosomes with known evolutionarily history for 320–350 MY, we addressed whether all centromeric repeats evolve at a similar pace, or if repeats at specific positions change rapidly. In the human genome, several centromeric monomers in

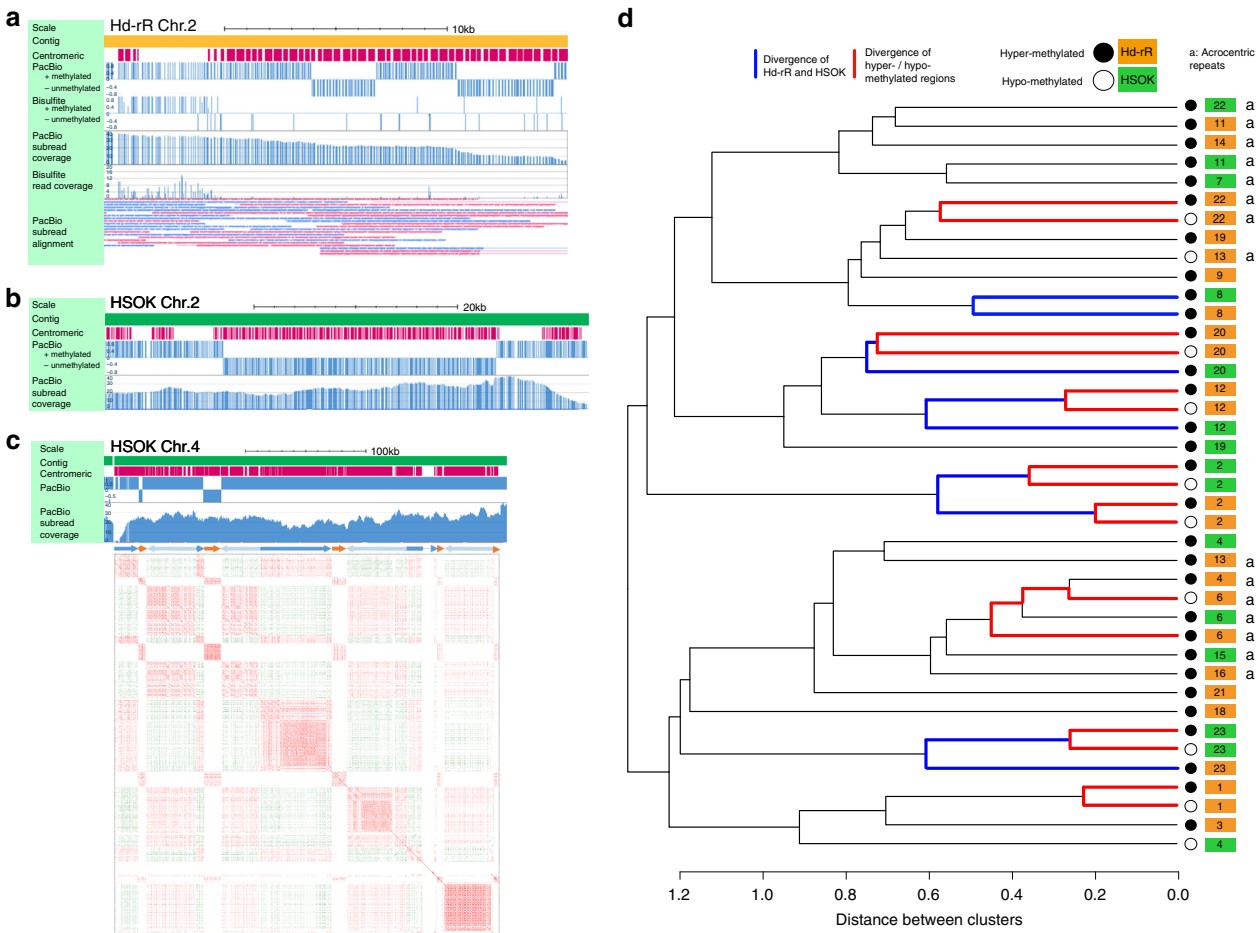

**Fig. 3** CpG methylation in centromeric repeats. **a** The tracks shown are, from the top, contigs layout as yellow bands, centromeric repeats as red bands, regional methylation prediction from PacBio reads (+, methylated; −, unmethylated), CpG-wise methylation from bisulfite reads, coverage of PacBio reads, coverage of bisulfite reads, and PacBio subreads alignments (red, forward; blue, reverse). Methylation calls by PacBio and bisulfite sequencing are inconsistent around the two unmethylated regions because most of bisulfite read coverages are very small (only 1) and are unreliable due to the repetitiveness of the centromeres. By contrast, PacBio reads achieved stable coverage over the repeat region. **b** A part of HSOK chromosome 2 with an unmethylated region that is syntenic to the region in Fig. 3a according to genetic markers (Fig. 1b). No bisulfite-treat short reads are available for the HSOK strain. **c** A ~305 Kbp centromeric repeat region in HSOK chromosome 4. The lower portion shows a dot plot of the region. Forward and reverse matches are colored red and green, respectively. Each dot represents 40-mer sequence match. Blue and orange arrows displayed above the dot plot show two patterns of centromeric repeats that do not match. A light blue arrow is inverse orientation of a blue arrow. **d** We clustered all hyper- and hypomethylated centromeric regions in Hd-rR and HSOK with at least 40 CpGs that we could reliably estimate from SMRT sequencing information (Methods, Supplementary Fig. 5). The respective orange and green boxes represent the Hd-rR and HSOK strains. The black and white circles illustrate hyper- and hypomethylated regions. Numbers indicate chromosome numbers. Black, blue, and red lines in the dendrogram respectively illustrate the timing of chromosome segregation, divergence of two strains (Hd-rR and HSOK), and divergence of hyper/hypomethylated regions in an identical chromosome of the same strain. Seven pairs of hypomethylated and hypermethylated regions (from top to bottom: Hd-rR chr. 22, 20, 12 HSOK chr. 2, Hd-rR chr. 2, HSOK chr. 23, Hd-rR chr. 1) are most similar to each other except for three exceptional cases (Hd-rR chr. 13, Hd-rR chr. 6, HSOK chr. 4). The rightmost column labels acrocentric repeats with "a"

acrocentric chromosomes are similar in sequence, and this is also true for metacentric chromosomes[40]. We grouped medaka chromosomes into acrocentric and non-acrocentric chromosomes (Fig. 2a). We then clustered monomers in centromeric repeats into clusters, and associated each cluster with the cluster of the highest similarity (Methods). We then categorized the closest-match associations between monomer clusters into three groups according to the positions of the monomers (Fig. 2b). We observed that centromeric monomers between non-acrocentric chromosomes were significantly less similar than those between acrocentric chromosomes (Fig. 2b, $p = 1.0 \times 10^{-6}$, Wilcoxon's rank sum test); therefore, non-acrocentric monomers evolved more rapidly than acrocentric monomers during speciation. These data suggest a greater role of non-acrocentric centromeres in genome diversity and speciation.

**CpG methylation in centromeric repeats**. Epigenetic mechanisms are known to play a crucial role in the establishment and maintenance of centromeres[41]. The CpG methylation status in centromeres has been examined using methyl-sensitive restriction enzymes[42], fluorescence antibody labeling[43,44], and bisulfite sequencing[45]. These studies showed that, on average, centromeric repeats were hypomethylated in core centromeres and were hypermethylated in pericentromeres in rice (Nipponbare)[43] and maize (Zea mays)[44]. Conversely, in mice (Mus musculus), the levels varied depending on tissue type, being higher for somatic cells, but intermediate and lower for sperm and oocytes, respectively[42,45,46]. However, these previous studies did not relate the methylation state of CpG sites with the structure of underlying centromeric repeats. We overcame this problem with our AgIn software to depict the global CpG methylation pattern over a

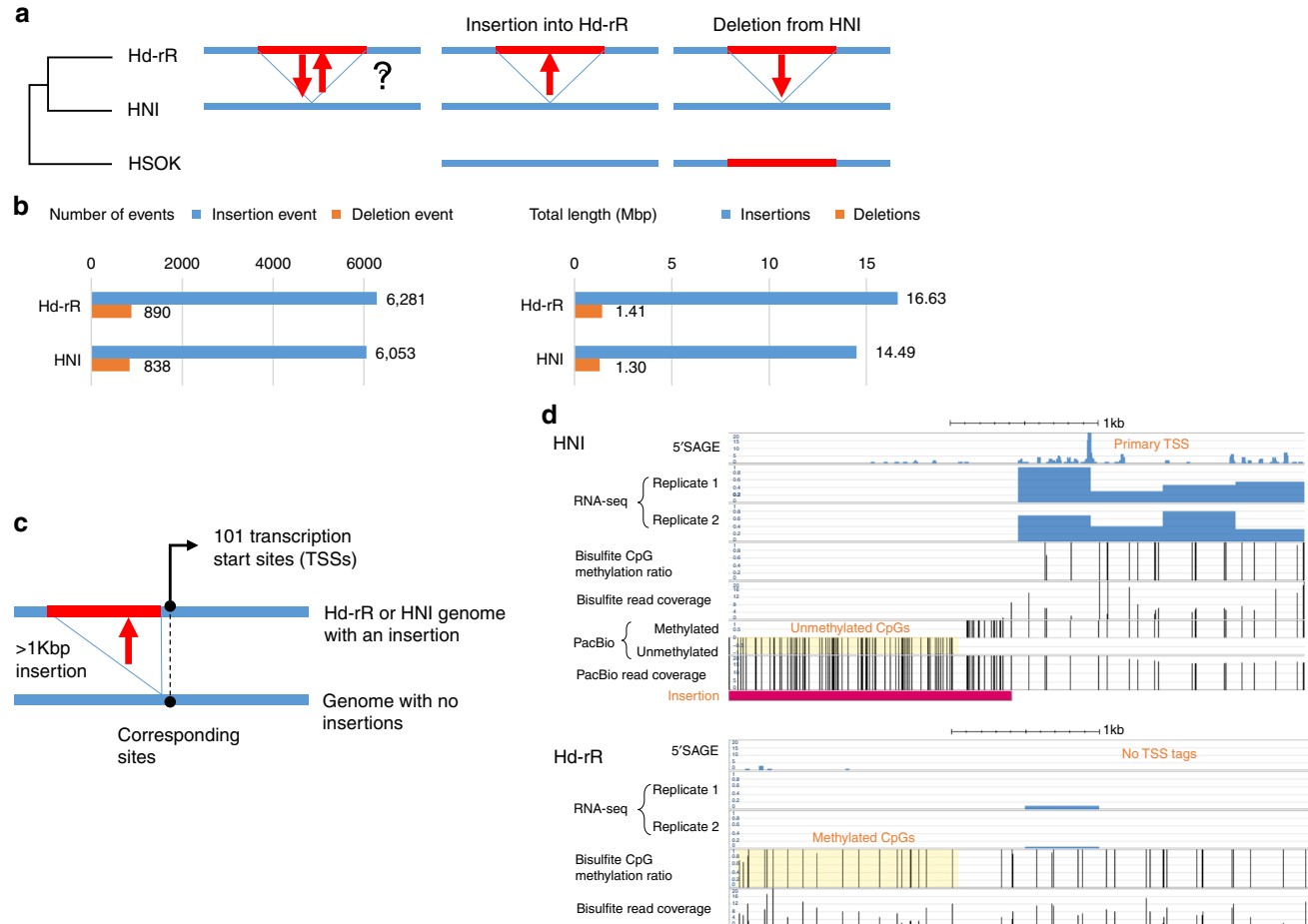

**Fig. 4** Mid-sized structural variants exhibit evolution toward longer genomes and genesis of genes. **a** We determined whether an NHEJ event was an insertion into, or a deletion from, one of Hd-rR and HNI, using HSOK as an outgroup for the two strains. **b** The left histogram shows the numbers of insertions and deletions into/from the two strains, whereas the right histogram shows the total lengths of insertions and deletions. Structural variants exhibit evolution toward longer genomes. **c** A schematic picture illustrating >1-kbp insertions into regions upstream of TSSs in one of the Hd-rR and HNI genomes. We identified 101 such occurrences (21 in Hd-rR and 80 in HNI) that had significantly increased transcript levels according to two biological replicates of RNA-seq experiments ($p < 1\%$, Wald test). **d** A pair of reciprocally best matching regions between the Hd-rR and HNI genomes that has a TSS with an insertion into the HNI genome that ranges from 842,236 b to 846235 b in contig 000284 F and corresponds to position 3415583 in contig 000015 F of the Hd-rR genome. The 5′SAGE track shows the frequency of 5′SAGE tags at each genomic position and highlights a highly expressed TSS in the HNI genome and no 5′SAGE tags in the Hd-rR genome. The RNA-seq tracks display normalized RNA-seq data (RPKM) in two biological replicates, labeled with 1 and 2, from early embryos (blastulae), supporting that the HNI TSS is highly expressed. The CpG methylation ratio at each CpG site was measured using bisulfite-treated short reads collected from blastulae. On the insertion upstream of the HNI TSS, no information on methylation ratios was available because the insertion was highly repetitive and has no bisulfite reads anchored on it; however, sufficient PacBio reads from testes could cover the insertion to determine that most of CpGs in it were unmethylated. Methylation levels of testes and blastulae are known to be highly correlated; for example, CpG sites downstream of the HNI TSS are methylated according to both bisulfite and PacBio data. In contrast, all CpG sites in the corresponding Hd-rR genome are methylated, which is reliably supported by sufficient read coverages

broad range of medaka centromeric repeats at fine resolution, including the boundaries of centromeres. Indeed, we reconfirmed this in centromeric repeats where bisulfite sequencing data were available, AgIn predicted bisulfite results at an accuracy of 88.7% on unmethylated CpGs and 90.7% on methylated CpGs (Methods). In non-centromeric regions, AgIn is capable of estimating methylation states of CpG sites with a high accuracy (sensitivity and precision of ~93.7%) from kinetic information of SMRT sequencing[47]; for example, Supplementary Fig. 4 shows typical examples of methylation states such that AgIn and bisulfite sequencing are concordant.

Adult medaka testes under reproductive laboratory conditions consist mainly of mature sperm and spermatogenic cells[48]. The centromeres obtained from these germ cells were found to be mostly hypermethylated (Supplementary Fig. 5), which

unexpectedly reflects the characteristics of somatic cells. We also reconfirmed this property by estimating the average methylation ratios of centromeres in testes and liver. Specifically, we aligned bisulfite-treated short reads from testes and liver[49] to the four representative centromeric monomers given in Supplementary Fig. 3b. The average methylation ratio in testes was 72.9%, which was close to 65.3%, the average in liver. However, we found that some centromeres contain hypomethylated domains. For example, Fig. 3a, b show two syntenic centromeric repeat regions with unmethylated sub-regions in chromosome 2 of Hd-rR and HSOK (see the genetic marker correspondence in Fig. 1b; dot plot in Supplementary Fig. 5). Figure 3c shows that HSOK chromosome 4 contained two hypomethylated regions which exhibited sequence similarity to each other. Similarly, we observed hypomethylated centromeric repeats in four Hd-rR and three

HSOK chromosomes (Supplementary Fig. 5). These examples showed diverged methylation patterns among centromeric repeats.

To understand this diversity, we analyzed underlying DNA sequences within centromeric repeats, and constructed a phylogenetic tree of centromeric repeats with distinct methylation status in terms of the sequence similarity calculated with spectrum kernel (Fig. 3d; Methods)[50,51]. Figure 3d shows the general tendency that the segregation of different chromosomes occurred first, followed by the separation of Hd-rR and HSOK ~25MYA (Fig. 1a). Afterwards, hypo/hypermethylated regions in individual chromosomes evolved independently and acquired unique sequence compositions that were not shared in common among different strains and chromosomes. This was confirmed by examining hypomethylated centromeric repeats in HSOK chromosomes, 2, 4 and 23 (Methods; Supplementary Notes; Supplementary Fig. 6). We remark two deviations from this general tendency. Centromeric repeats in acrocentric chromosome are more similar in sequence than those in non-acrocentric chromosomes are, suggesting exchanges of repeats between acrocentric chromosomes. For example, the hypomethylated Hd-rR chr. 6 and the hypermethylated Hd-rR chr. 4 were reciprocally most similar to each other, and they might be exchanged in Hd-rR. Hypomethylated repeats in HSOK chromosome 4 (orange repeats in Fig. 3c) are more similar to repeats in Hd-rR chromosome 1 than to repeats in chromosome 4, suggesting they might jump in HSOK chromosome 4 from another chromosome. Overall, DNA methylation patterns in centromeres were not correlated with centromere sequence phylogeny, but hypo/hypermethylated regions in each chromosome evolved independently.

**An evolutionary tendency toward longer genomes**. Comparisons among the contigs of the three inbred strains revealed substantial numbers of large SVs including insertions, deletions, duplications, and inversions. The biggest SV is a >15 Mbp inversion in chromosome 11 (Supplementary Figs. 2, 7), which was suggested[52] but unclear based on the prior Sanger-sequence genome assembly[19]. Mid-sized (1–50kbp) SVs are more frequent and known to have various impacts on genome function[14], but have been difficult to identify using short reads[53]. Thus, we enumerated mid-sized SVs in the Hd-rR and HNI genomes (see SV positions in Supplementary Tables 14, 15, 16), and found that 96.9% were either insertions or deletions (Fig. 4a, Supplementary Fig. 7d, Supplementary Notes). Remarkably, after Hd-rR and HNI diverged ~18 MYA[20], insertions into either strain were more prevalent than deletions, by a factor of seven (Fig. 4b). In total, the difference between the total lengths of insertions and deletions was ~15 Mbp in both Hd-rR and HNI, evidencing a tendency towards longer genomes during local evolution, i.e. speciation.

**Impact of large-scale insertions on transcription regulation**. Inserted DNA fragments in each genome may affect gene regulation during speciation. In the Hd-rR and HNI genomes, we first identified TSSs supported by 5′SAGE tags[19,54] that had 1−10-kbp mid-sized insertions within 100 bp from the TSSs in one of the genomes, and we then detected 101 differentially expressed TSSs between Hd-rR and HNI. Specifically, from each strain, we generated two RNA-seq biological replicates from early embryos (blastulae), and we sequenced and processed these four data sets using DESeq2[55] to detect the 101 TSSs that had insertions in their upstream regions (Supplementary Fig. 10a) and had increased transcript levels at a stringent statistical significance ($p < 0.01$, Wald test) (Fig. 4c, d). These insertions were significantly

correlated with increases in the average GC ratio ($p < 10^{-6}$), increases in the average CpG ratios ($p < 10^{-3}$), and decreases in the CpG methylation levels upstream of TSSs in blastulae ($p = 0.39$; Supplementary Fig. 10b; Methods). We could determine CpG methylation states for 92 of the 101 TSSs using short bisulfite-treated reads from blastulae; however, for the remaining 9 TSSs, we could not. Figure 4d illustrates a difficult situation where we could not determine hypomethylation of a highly repetitive insertion using short reads, but we could with long PacBio reads from testes. Although testes and blastulae are different tissue types, their CpG methylation levels were known to be highly correlated ($R^2 = 0.73$) in Hd-rR[49].

Among these 101 increased transcripts, 29 transcripts were undetectable by the two RNA-seq biological replicates from the counterpart region in the other strain, suggesting that these transcripts were produced from newly generated TSSs. Searching 1000-bp regions downstream of those 101 TSSs for predicted genes revealed only 9 coded predicted genes, suggesting that transcripts from the remaining 92 regions were non-coding. In particular, 27 of the above 29 novel transcripts (93.1%) were non-coding. In summary, we identified 101 insertions that increased the GC ratio and CpG ratios, and lowered DNA methylation levels, leading to significantly elevated transcriptional expression or de novo transcription.

## Discussion

In the present study, we aimed to understand how centromeres and large-scale SVs evolve and contribute to genome diversity during vertebrate speciation. To this end, we generated long contigs for three inbred medaka strains with twenty-two centromeric repeat regions of length 20–345 kbp. We found that non-acrocentric monomers evolved more rapidly than acrocentric monomers, yielding hypomethylated regions with distinct sequence compositions. The apparent slow evolution of acrocentric centromeres might be accounted for by the meiotic telomere bouquet, to which telomeres are attached during meiosis (Supplementary Fig. 11). The telomere bouquet has been observed in yeasts and plants and brings centromeres of acrocentric chromosomes into close proximity[56]. We speculate that it facilitates more frequent exchanges of centromeric repeats between acrocentric chromosomes than between non-acrocentric ones, thereby resulting in faster centromere sequence evolution in non-acrocentric chromosomes. Less frequently, however, exchanges between acrocentric and non-acrocentric repeats were also suggested in our study. Collectively, our data show, for the first time, the global view of centromere interaction between particular chromosomes during vertebrate evolution.

Our study is also the first to reveal the specific pattern of hypomethylated and hypermethylated domains in centromeric repeats, which has been overlooked by traditional approaches. Analysis of underlying DNA sequence showed that the variation of non-acrocentric CpG methylation occurred after the divergence of two medaka strains (Hd-rR and HSOK), demonstrating that centromeres accumulated epigenetic diversity as well as the sequence diversity during speciation. Although centromere identity is known to be primarily defined by the epigenetic specification, in particular, by the presence of the histone H3 variant CenH3/CENP-A[57], a specific pattern of CpG methylation could play some roles in centromere evolution through meiotic centromere pairing.

We observed that each local strain has independently experienced thousands of mid-sized insertion events. However, those insertions have not yet caused reproductive isolation, as Hd-rR and HNI can produce fertile hybrid offspring. Since the two strains equally increased their genomes to equal degrees by these

mid-sized insertions, we speculate that increases in genome size might be a general tendency for vertebrate genomes undergoing local evolution under natural conditions. In addition to genome size, it has been speculated that transposable elements (TEs) broadly contribute to diversity in gene regulation[58] and the genesis of novel genes with new functions in eukaryotic genomes[59]. In this context, of particular interest are 101 genomic positions at which mid-sized insertions could participate in the regulation of genes, and many of these insertions are likely to be TEs (Supplementary Notes; Supplementary Fig. 10c). Such insertions significantly increased the GC content, CpG ratio, and extent of CpG unmethylation, thereby increasing gene transcription when they were inserted upstream of preexisting TSSs. More importantly, 29 insertions appeared to generate strain-specific transcripts, and 27 (>90%) were found to be non-coding. The function of these non-coding RNAs and proteins is yet to be determined, but some may contribute to phenotypic variation between the two strains, along with genes upregulated by insertions, leading to speciation[24]. In general, TE-mediated novel transcripts are usually non-functional but can be co-opted into novel regulatory circuits during speciation and evolution[59]. Thus, our data highlight the importance of mid-sized insertions in the process of vertebrate speciation. Further analysis of the mid-sized insertions associated with novel transcripts and increased transcription will provide important clues to the genomic basis for vertebrate speciation.

## Methods

**Preparation of genomic DNA and SMRT sequencing.** Genomic DNAs from the three inbred medaka strains were used to prepare SMRTbell libraries. DNA was sheared, using a g-TUBE device (Covaris Inc., Woburn, MA, USA) operating at 4,300 rpm and purified using a 0.45 × volume ratio of AMpure beads (Pacific Biosciences, Menlo Park, CA, USA). SMRTbell libraries for sequencing were prepared using the "20 kb Template Preparation using BluePippin Size Selection System (15 kb Size Cutoff)" protocol. Briefly, this features (1) DNA repair; (2) blunt ligation with hairpin adapters employing the SMRTbell template Prep Kit 1.0 (Pacific Biosciences); (3) size selection using the BluePippin DNA size selection system of Sage Science; and (4) binding of DNA fragments to Polymerase P6 using the DNA Sequencing Reagent 4.0 (Pacific Biosciences). SMRTbell libraries were sequenced on a SMRT Cell (Pacific Biosciences) using magnetic bead loading and P6-C4/P5-C3/P4-C2 chemistry. Sequence data were collected with the aid of a magnetic bead collection protocol. The insert size was 20 kb; "stage start" was enabled, and 240-min movies were run employing PacBio RS Remote. Primary filtering was performed on the PacBio RS II Blade Center server.

**Correcting assembled contigs using Illumina reads.** After polishing assembled contigs using Quiver[6], we sought further improvements by correcting sequencing errors in single nucleotides and short indels; we aligned short Illumina reads to the contigs using Pilon[25]. The numbers of corrected small deletions and insertions were higher than those of corrected single bases in the HNI and HSOK strains (Supplementary Table 18). In the Hd-rR assembly, 97.90% of 10-kbp non-overlapping regions exhibited sequencing error rates of <0.1%, whereas the remaining regions of total length ~16.6 Mb (2.10%) had error rates of >0.1% (Supplementary Table 19). The latter regions were often clustered consecutively among contigs. Of these regions, sub-regions of a total length of ~231 kb exhibited remarkably high read coverages on both short and long reads, and they were indeed centromeric repeats (Supplementary Table 20). Because distinct repetitive regions had failed to become separated, being somewhat merged into each of the sub-regions, we did not correct these regions using short Illumina reads.

**Generating a chromosome map for each strain.** We used 2,347 SNP genetic markers to anchor contigs of the three strains to the 24 medaka chromosomes (Fig. 1b) using the alignment software program ispcr (in silico PCR), which is available at https://github.com/mkasa/klab/blob/master/script/ispcr. We ordered the contigs along each chromosome according to the genetic distances between markers. Some contigs were subsumed by other (longer) contigs; we eliminated the former redundant contigs. We detected 17 misassembled contigs in the Hd-rR strain, 16 in the HNI strain, and 8 in the HSOK strain; all contained genetic markers originating from two different chromosomes. We corrected these misassembled contigs by dividing them into two subcontigs by reference to the genetic markers, and anchored the partitioned (sub)contigs to their respective chromosomes. We also anchored remaining Hd-rR contigs that were connected by multiple BAC/fosmid-end pairs. Specifically, after considering the estimated median

sizes of BAC and fosmid clones (135 kbp and 37.5kbp), we used BAC-end (fosmid-end) reads mapping to a position within 150 and 50kbp from one end of a contig (Supplementary Fig. 1). In contrast, for HNI and HSOK, sufficient BAC-end and fosmid-end pairs were unavailable and no Hi-C data were collected. We instead located 44 HNI contigs with no genetic markers to chromosomes by reference to their best matches to Hd-rR contigs. Some Hd-rR, HNI, and HSOK contigs remain unoriented because they were associated with only a single genetic marker, or multiple genetic markers at the same genetic distance apart. We attempted to determine the orientation of each unoriented contig by reference to the orientations of the best-matched contigs in the other strains (Supplementary Fig. 2).

**Assembly by Hi-C data.** We used Hi-C data to locate 11 orphan contigs which contained centromeric repeats but failed to be anchored onto chromosomes because of the absence of genetic markers on them. First we trained a naive Bayes classifier to predict the chromosome of each orphan contig considering its contact frequency information with individual chromosomes. For each orphan contig, contact frequency $a_i$ with chromosome $i$ was calculated by the number of Hi-C reads mapped between the contig and chromosome $i$. The contact frequency variables $a_1,\ldots,a_{24}$ are conditionally independent of each other given the chromosome $i$. The posterior probability of the orphan contig anchored to chromosome $c$ is

$$p(c|a_1, \ldots a_{24}) = \frac{p(c) \prod_{i=1}^{24} p(a_i|c)}{Z}$$

where $p(c)$ is a prior probability proportional to the number of contigs in chromosome $c$, $p(a_i|c)$ is a conditional probability of contact frequency $a_i$ under the condition that the orphan contig was anchored to chromosome $c$ and $Z$ is a normalization factor. We verified the correctness of the above naive Bayes classifier by checking whether 500 contigs that were already anchored by genetic markers were also accurately classified to chromosomes, which had the highest posterior probability. Indeed, we confirmed that all contigs could be correctly classified. Thus, we assigned the chromosomes to 11 orphan contigs with centromeric repeats by using the naive Bayes classifier.

Next we predicted the precise positions and orderings of the eleven orphan contigs in their assigned chromosomes. To this end, we utilized the property that, along each chromosome, the contact frequency increased almost exponentially towards one position (Supplementary Fig. 8). Certainly, the average contact frequency of the 1 Mbp region surrounding the position was clearly higher than that outside. According to this property, for each orphan contig that was anchored by the naïve Bayes classifier, we calculated the contact frequency between the orphan contig and anchored contigs in the chromosome assigned to the orphan contig, and located the orphan contig next to the position, which had the highest contact frequency.

**RNA-sequencing and gene annotations.** Strand-specific paired-end RNA-Seq data were collected from 57 tissue types using Illumina Hiseq4000 (Supplementary Table 9). We assembled the RNA-Seq data in individual 57 tissue types separately using Trinity with option "–SS_lib_type RF" to process strand-specific paired-end data properly. We obtained 7,325,838 assembled contigs, and their average length was 1055.51 bp (Supplementary Table 9). We then merged the assembled RNA-Seq contigs from all the tissue types into one set so as to predict a set of medaka genes. We ran MAKER2[30] gene annotation pipeline twice, following the standard procedure[60], and produced SNAP[61] HMMs from a set of fundamental genes that we identified using CEGMA[62]. To complement SNAP HMMS, we also generated GeneMark HMMs by running another gene finder GeneMark-ES[63]. We then performed the first application of MAKER2 using the SNAP HMMs, GeneMark-ES HMMs, and assembled RNA-Seq contigs on genomic contigs of ~20Mbp in size, and we used the MAKER2 output to revise SNAP HMMs. Subsequently, we did the second application of MAKER2 with the revised SNAP HMMs, GeneMark HMMs, and the assembled RNA-Seq contigs on all the genomic contigs (version 2.2.4). Afterwards, we selected gene annotations on anchored genomic contigs. As MAKER2 outputs alternative splice genes with AED (annotation edit distance) scores, we selected a gene with the smallest AED score as the representative from each locus. We obtained 93,896 putative genes on anchored contigs including a reliable set of 29,267 genes with 5′ and/or 3′ UTRs (Supplementary Table 10). To assess the new set of representative genes, we mapped 24,674 protein sequences in the set of genes predicted from the medaka version 1 to the newly identified representation genes using BLASTP. A total of 94.1% of the alignments met the condition that e-values were <1E−20 and alignment lengths were at least 50% the protein sequences, thereby confirming the new set of genes include most of the previous set.

**Clustering monomers.** Monomers in each chromosome of each strain separately were clustered using DNACLUST[64]. From each cluster with >10 monomers, the longest monomer was selected as the representative. To calculate the alignment between a pair of two representative monomers, the similarity between the pair of monomers was defined as

(number of matched bases)/(length of the shorter monomer),

and the distance between the pair of monomers was defined as (1−similarity). Supplementary Table 21 shows the strain, chromosome, and number of monomers in each monomer group (cluster), and the similarities of all pairs of representative monomers. We associated each monomer cluster, X, with the best matching cluster, Y, whose representative monomer had the highest similarity with the representative of X. The highest similarity is useful in approximating the time when the two monomer clusters might have exchanged monomers.

According to the distance defined above, we generated a hierarchical clustering of representative monomers using hclust with the ward.D2 method in R software (Supplementary Fig. 3a). The clustering showed four groups of representative monomers named SF (Suprachromosomal Family)[65]. A representative was selected from each of the four groups. We examined if the representative monomer identified by Melters et al.[36] matched centromeric repeats in all Hd-rR chromosomes with a high identity; however, it matched centromeric repeats in only seven chromosomes (chr. 4, 6, 11, 13, 14, 16, and 21) with a mean identity of >85% (Supplementary Fig. 3c). Six of these seven chromosomes were in SF2, and therefore the monomer identified by Melters et al. was used as the representative of SF2. To select a representative monomer for each of the remaining three groups, we decomposed centromeric repeats into monomers using RepeatMasker, aligned individual monomers to the original centromeric repeats using BLAST, and selected the optimal monomer with the best score as a representative for each chromosome, where the score was defined as the sum of

$$(\text{alignment identity} * \text{alignment length} / \text{query monomer length})$$

over all hits. The respective representative monomers in chromosomes 9, 12 and 2 were the representatives of SF1, SF3, and SF4. Centromeric monomers in chromosomes of a group matched the representative of the group with a high identity (Supplementary Fig. 3c).

**Designing centromere-specific DNA probes**. The sequences of the four representatives in Supplementary Fig. 3b were used as centromere-specific DNA probes for our fluorescence in situ hybridization experiment.

**Fluorescence in situ hybridization experiment**. Centromeric satellite DNA was synthesized by annealing and extending two DNA oligos using TaKaRa ExTaq (TaKaRa), followed by subcloning into pCR™II-TOPO®vector(Thermo). DNA probes were prepared by cutting and labeling the plasmid DNA with biotin, using the Nick Translation Kit (Roche). Medaka fibroblast cells were treated with 0.05 µg/ml of corcemid (for probe1,2) or 1 µM of nocodazole (for probe3, 4, all) for 4–5 h. After trypsinization, cells were hypotonically swollen in 75 mM KCl for 20 min, fixed with ice-cold Carnoy's solution (1:3 acetic acid: methanol), then spread onto slides. After RNase treatment and denaturation of chromosomal DNA, hybridization was carried out by dropping probe DNA solution onto slides and incubating at 37 °C for overnight. After washing, chromosomal DNA was incubated with avidin-FITC (Vector Laboratories) for 1 h. After the final wash, slides were mounted with Vectashield Plus DAPI (Vector Laboratories). Images were acquired using a fluorescence microscope (LSM710; Zeiss).

**Searching for centromere and telomere regions in chromosomes**. The ratio of centromeric satellites in the entire genome was quantified by searching raw PacBio subreads for a representative centromeric monomer sequence identified by Melters et al.[36] using RepeatMasker (version 4.0.6, http://www.repeatmasker.org) (Supplementary Table 11). Subreads was filtered according to the following criteria: the subread length was >1 kb, and the average quality value (QV) was >10. We then calculated the genomic fraction of centromeric satellites as the ratio of the total amount of centromeric satellites in the filtered subreads to the total length of the filtered subreads. The assembled genomes was searched for the representative centromeric satellite monomer with RepeatMasker (Supplementary Table 12).

To validate the correctness of the sequence assembly at centromeric regions, raw PacBio subreads were mapped to the centromeric regions using BLASR (version 5.2.6fa6cc2), and we used those anchored subreads such that the alignment length was >5 kb, the overall alignment identity was >80%, and the alignment identity of 1 kb subsequences at both ends of the alignment was >85%. Using a genome browser, we visualized the alignments of raw subreads as well as satellite arrays using the output of RepeatMasker (Supplementary Fig. 5). Individual centromeric regions were inspected and confirmed that the regions were covered by overlapping, non-redundant raw subreads that started from and ended at different genomic positions. To compare centromeric monomers, pairwise alignment of monomer sequences was done with EMBOSS needle (version 6.5.7)[66].

To detect telomeric repeats, we first enumerated all possible repetitive elements in assembled contigs using Tandem Repeats Finder (version 4.0.9), and we selected such repeats that their units are TTAGGG, the sequence of nucleotides present in vertebrate telomeres (Supplementary Table 17).

**Methylation calls using SMRT and bisulfite sequencing**. Methylation call from SMRT long reads was performed using AgIn algorithm, which is detailed in what follows[47]. For methylation analysis, we used SMRT reads sequenced with P6-C4

chemistry and avoided mixing reads from different polymerase and chemistry, which is not guaranteed to produce reliable result. Mapping and generation of modification summary (modifications.csv) were performed using SMRT Pipe with its default settings for the general resequencing protocol. The result was then processed by AgIn algorithm[47] to extract a set of hypomethylated regions. Specifically, we used the same parameters tuned for P6-C4 (beta for P6-C4 and gamma = −0.55), and set the minimum number of CpGs in each predicted region to 40. Bisulfite-treated short reads were downloaded from SRA (Accession No. SRX149585) and were processed by Bismark[67] to perform genome conversion, mapping of reads to converted genome, and production of methylation summary as bedGraph file. To align reads with bowtie2, we used the parameters: -L 32 -N 0 −ignore-quals. Each CpG site was classified as methylated if the strict majority of the mapped reads supported that it was methylated, otherwise as unmethylated. During the calculation of consistency between the results of AgIn and bisulfite sequencing, we considered CpG sites with bisulfite read coverage ranging from 2 to 9, in order to exclude positions with an abnormally high coverage, which were likely to have identical copies in the genome. Among CpGs within the hypomethylated (hypermethylated, respectively) regions in centromeric repeats that we estimated from PacBio reads, 88.7 % (90.7%) were called as unmethylated (methylated) from bisulfite reads. Therefore, each technology supported the methylation calls from the other when methylation information is available from both.

We also calculated the average methylation ratios in centromeres in testes and liver by using bisulfite-treated reads collected from testes and liver[49], and by aligning the reads to the four representative monomers in Supplementary Fig. 3b. The average methylation ratio in testes was 72.9%, which was close to 65.3%, the average ratio in liver. Specifically, the respective numbers of methylated and unmethylated cytosines in liver were 20,245 and 10,827, which yielded the average 72.9% (=20,245/(20,245 + 10,827)), while those in testes were 19,103 and 7,356, and hence the average was 65.3% (=19,103/(19,103 + 7,356)).

**Phylogeny of hyper-/hypomethylated centromeric regions**. For the analysis of evolution of CpG methylation in centromeric repeats, we used all Hd-rR or HSOK chromosomes that had either hyper- or hypomethylated centromeric repeat regions. Let **A** and **B** denote the normalized vector of k-mer frequencies in repeat regions, A and B, respectively such that $\|\mathbf{A}\|^2 = \|\mathbf{B}\|^2 = 1$. To perform cluster analysis, we defined the distance between regions, A and B, by $D(\mathbf{A},\mathbf{B}) = \sqrt{\| \mathbf{A} - \mathbf{B} \|^2}$. The formula is then transformed to

$$\sqrt{\| \mathbf{A} \|^2 + \| \mathbf{B} \|^2 - 2K(\mathbf{A},\mathbf{B})} = \sqrt{2 - 2K(\mathbf{A},\mathbf{B})},$$

where $K(\mathbf{A},\mathbf{B})$ denote the inner product of **A** and **B** that represents a sequence similarity between repeat regions, A and B, which is equivalent to the k-spectrum kernel[50], a widely used measure in sequence comparison. Based on these pairwise distance, we generated a hierarchical clustering of the regions with the UPGMA method[68]. In our analysis, we set k to 8 in Fig. 3d because the setting could separate the segregation of chromosomes and the divergence of the medaka strains in the clustering. We calculated spectrum kernel, clustering, and final visualization using R statistical environment (https://www.R-project.org/), and especially, the "kebabs" package for kernel-based analysis[69].

**SVM analysis of hyper-/hypomethylated centromeric regions**. We attempted to characterize sequence composition in hyper-/hypomethylated centromeric regions using support vector machine (SVM) with k-spectrum kernel[51]. For this analyses, we used the HSOK genome chromosomes 2, 4, and 23 with >10kbp hypomethylated and >10kbp hypermethylated regions that were sufficiently long to perform a reliable analysis on sequence compositions. The positions and lengths of hypomethylated centromeric regions are chr2: 10,434,969−10,459,620 (of length 24,652), chr4: 12,757,332−12,759,630 (2,299), chr4: 12,811,931−12,825,347 (13,417), chr23: 8,953,289−8,962,353 (9,065), chr23: 8,963,500−8,968,835 (5,336), while hypermethylated regions are chr2: 10,501,805−10,506,125 (of length 4,321), chr2: 10,510,445−10,525,439 (14,995), chr4: 12,739,300−12,755,955 (16,565), chr4: 12,761,969−12,810,081 (48,113), chr4: 12,825,809−12,992,351 (166,543), chr4: 13,003,454−13,053,416 (49,963), chr23: 8,948,126−8,953,059 (4,934), chr23: 8,969,236−8,975,202 (5,967).

These sequences were divided into 200-bp non-overlapping sub-regions that were used as the training and test data. To examine whether a k-mer SVM can discriminate hypomethylated domains (HMDs) and hypermethylated domains in centromeric regions, we performed a five-fold cross validation; namely, we partitioned the data set into five subsets, used four data sets to train a k-mer SVM, tested the other subset to test the accuracy of the SVM, and repeated this five times to calculate the average accuracy. We also checked whether some relevant k-mers were shared in common among different chromosomes. We used the sequences on two of the three HSOK chromosomes (chr2, 4, and 23) as training data and the sequences on the remaining chromosome as test data. To identify relevant k-mers that underlie in HMDs in non-centromeric regions, we used DNA sequences from HMDs and methylated regions in Hd-rR blastula embryos and used as training data, where HMDs were defined in the previous paper[70].

**Comparing insertions into or deletions from a genomes**. We enumerated structural variants (SVs) between two genomes according to the following steps: (1) used generic markers to identify reciprocally best matching pairs of contigs in the genomes, (2) listed local alignments between the genomes using LAST, (3) chained the local alignments using the idea of dynamic programming, and (4) categorized SVs into NHEJ, NAHR, and inversions by analyzing the boundaries of SVs. We publicized the data processing pipeline called PBEC at http://pbec.gi.k.u-tokyo.ac.jp/. After comparing Hd-rR and HNI genomes, we obtained a set of candidate mid-sized insertion or deletion events; however, an outgroup genome was required to determine whether a given event was an insertion into, or a deletion from, the focal genome (Supplementary Fig. 9a). We used HSOK as an outgroup; we mapped (to HSOK) the two 2,500-bp regions upstream and downstream from the positions in HNI (or Hd-rR, respectively) where the insertion/deletion events had occurred in Hd-rR (HNI). Supplementary Fig. 9a illustrates the procedure. We measured the distance between the alignments of the two 2,500-bp regions in the HSOK genome. Supplementary Fig. 9b shows the frequency distribution of these distances, and exhibits two peaks around 0 and >1000. As the two peaks were thus clearly separated, we classified events using the heuristic whereby events in the peak around 0 were insertions, and those around the other peak were deletions. The peak around 0 is broad because the three strains collected mutations during evolution. Supplementary Fig. 9c shows the frequency distribution of lengths of insertions.

**Significances of key parameters around TSSs with insertions**. We identified 1,213 reciprocally best-matched pairs of regions in the Hd-rR and HNI genomes; one such region genome had a TSS site supported by 5′SAGE tags and a 1–10 kbp insertion within 100 bp of the TSS, but the other had no such insertion. Specifically, the respective Hd-rR and HNI genomes had 407 and 806 regions with insertions. To identify genes that were differentially expressed between Hd-rR and HNI, we generated two biological replicates of RNA-seq experiments from early embryos (blastulae) of the two strains. Specifically, we eliminated ribosomal RNA using the RiboMinus Eukaryote Kit v2 (Life Technologies), and collected 50 bp single-end stranded reads using Illumina HiSeq1500 and the KAPA stranded RNA-seq Library Preparation Kit. We obtained 57.2 M and 56.5 M reads from two biological replicates of Hd-rR, and 62.1 M, and 73.1 M reads from HNI. Of these, 43.0 M and 42.9 M reads were aligned to the Hd-rR genome, and 48.0 M and 55.8 M to the HNI genome, respectively. We then processed the four data sets using DESeq2[55] and detected 101 TSSs (21 in Hd-rR and 80 in HNI) that had 1–10 kbp insertions and had more transcription levels than the corresponding TSSs in the other strain at a stringent statistical significance ($p < 1\%$, Wald test).

These insertions were categorized into classes of transposable elements by RepeatMasker 4.0.6 according to the criterion that >30% of each insertion matched a transposable element. Supplementary Fig. 10a shows the frequency distribution of distances between TSSs and insertions. For the regions with no insertions, we calculated the pseudo TSS positions corresponding to the TSSs of the other regions if TSSs were absent. For each region, we next calculated the GC content ratio, CpG ratio, and ratio of unmethylated cytosines to CpG sites in the 500-bp region upstream of the (pseudo-) TSS. We explored the statistical significance of the increase in each parameter, between regions with and without insertions, using the one-sided Wilcoxon signed rank test. These insertions correlated with increases in the average GC ratio (either C or G) upstream of TSSs ($p < 10^{-6}$; Supplementary Fig. 10b) presumably because the average GC ratio of all insertions, 41.8%, was significantly higher than those of the entire Hd-rR and HNI genomes, 40.85% and 40.63%, respectively. Similarly, the respective CpG ratios upstream of TSSs with and without insertions were 3.19% and 2.01%, respectively, and this difference was significant ($p < 10^{-3}$). We also examined the CpG methylation levels in 500 bp regions upstream of the 101 TSSs with insertions using bisulfite-treated reads from medaka blastulae of early embryos[49]. In 92 of the 101 TSSs, we observed that CpG methylation levels were significantly lower than the levels of those lacking insertions ($p = 3.9\%$) as illustrated in Fig. 4d. For the remaining 9 TSSs, we could not measure the methylation levels reliably using short bisulfite reads due to highly repetitive insertions as illustrated in Fig. 4d.

**Ethics approval**. All experimental procedures and animal care were carried out according to the animal ethics committee of the University of Tokyo (Approval No. 14–05).

**Data availability**. We deposited the sequence data of SMRT reads and assembled genomes from Hd-rR, HNI, and HSOK at the NCBI SRA (BioProject Accession: PRJNA325079 for Hd-rR, PRJNA325193 for HNI, PRJNA325194 for HSOK), and the in situ Hi-C reads from Hd-rR and d-rR at NCBI SRA (PRJNA378460 for Hd-rR, PRJNA378464 for d-rR). The accession number of the RNA-seq data for gene prediction is DRA005309, and the accession number of two RNA-seq biological replicates from blastulae of Hd-rR and HNI is SRP116580. The assembled genomes of the three strains, a comparative genomic analysis of the three strains, a medaka gene model, DNA methylation estimation from SMRT sequencing kinetic data, and a web browser for visualizing these datasets are available at http://utgenome.org/medaka_v2/.

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

## Acknowledgements

This study was supported in part by the Core Research for Evolutional Science and Technology (CREST) program (JPMJCR13W3) from Japan Science and Technology Agency (JST) to H.T. and S.M., and a Grant-in-Aid for Japan Society for the Promotion of Science (JSPS) Fellows (15J03645) to Y.S. We thank Yoichiro Nakatani, Shoichiro Oishi, Junko Taniguchi, Masahiro Kasahara, Takamasa Imai, and Yuichi Motai.

## Author contributions

K.I., S.T, Y.S., R.N. and K.D. are equally contributing first authors. S.M. and H.T. designed and supervised the research and wrote the paper; R.N. and M.K. collected DNA samples and Hi-C data; K.I. performed the genome assembly with J.Y.; K.I. constructed the chromosome map, positioned contigs with centromeric repeats, and analyzed the Tol2 and Y-specific regions; S.T., S.M., K.I. and J.Y. analyzed centromere DNA evolution, Y.S. CpG methylation and its evolution in centromeres, and R.N. sequence compositions in hypomethylated centromeric repeats; J.Y. set up computational facilities; R.N. and Y.I. performed FISH experiments; R.N., Y.U. and N.I. collected RNA-seq data; K.D. constructed the medaka gene model with N.I. and analyzed large insertions.

## Additional information

**Competing interests:** The authors declare no competing financial interests.

