## [Peer Review File · Nature Communications]

Reviewers' comments:

Reviewer #1 (Remarks to the Author):

The manuscript submitted to Nat. Communication was moderately revised from a previous submission to Nature. The authors have now clarified that data have been deposited to public repositories. Certain sections have been rewritten and the overall clarity of presentation was improved. However, the revision failed to address certain major concerns of the study. For example, one major problem raised by this reviewer is that biological replicates for RNA-seq experiment are absolutely required for calling differentially expressed genes and can not be replaced by technical replicates (see comment 1 below). In addition, critical information is missing for a newly added analysis Fig. 3D, which led this reviewer to question whether the result was representative. In summary, although the manuscript was moderately improved from its previous version, to this reviewer significant revisions are still needed before the manuscript can be published.

1. Regarding this reviewer's comment that the manuscript does not have biological replicates for identifying many (457 out of 716) insertions near the TSS leading to increase of transcription activity. The authors have answered that 'we did not have biological replicates, we generated technical replicates and used DEseq2 to verify the statistical significance of transcription alternation'. The answer to this comment is not acceptable since technically it is fully feasible to repeat the experiment and carry out RNA-seq with biological replicates for early embryos of Hd-rR and HNI.
2. The last section of the study, TSS associated with insertions were examined for CpG methylation and gene expression. However, DNA methylation and gene expression was profiled using different tissues, with the former using germ cells and later using early embryo. The authors need to state this major caveat (tissue mismatch) explicitly in the text, since the decrease in CpG methylation and gene activation may not be correlated when the two features were analyzed using matched tissue.
3. The manuscript states 'The centromeres obtained from these germ cells were found to be mostly hypermethylated, which unexpectedly reflects the characteristics of somatic cells.' However, without presenting any medaka somatic cell methylome in the study, it is unclear how the authors can reach the conclusion that germ cell methylomes are similar to somatic cells.
4. Why does Fig. 3D only include a subset of chromosomes?
5. The centromeric regions chosen for analysis in Fig. 3D were not justified. What criterion were used for selecting the chromosomes and regions for the analysis? It is unclear how representative these regions are. The length of tested regions ranged from 2.3kb to 166.4kb. What percentage of assembled medaka centromeres were included in this analysis?
6. The manuscript should explicitly state a conclusion of the analysis shown in Fig. 3D. It seems centromere sequence correlated with chromosomes, and acrocentric vs. non-acrocentric centromeres. What is the conclusion regarding DNA methylation? Is DNA methylation correlated or not correlated with centromere sequence phylogeny?
7. To "Impact of large-scale insertions...", Figure 4D and methods - I question the finding that "methylation levels upstream of TSSs with insertions were significantly lower than the levels of those lacking insertions". In fact, the caption to Figure 4D provides the reason for why I believe this statement is not correct: "methylation ratios are not available for the insert because of high repetitiveness". Did the analysis pipeline for methylation levels account for this? If no

Illumina bisulfite reads can be anchored, what is the methylation result from the PacBio reads that assembled this region/insertion in the first place?

8. Figure 3 – A major issue in the first version of this manuscript was the accurate comparison and representation of bisulfite data and SMRT kinetic analysis; While the figures have improved, they are still not convincing because of the low mapping rate of bisulfite reads to the centromeres. The authors should include one panel showing a representative region from a chromosome arm that is highly covered with bisulfite reads and clearly shows congruence to SMRT kinetics.

9. Figure 2 is not very meaningful and could be placed entirely into the supplement.

Reviewer #2 (Remarks to the Author):

The authors have improved the paper and answered my concerns. I like the presentation of the methylation in centromeres and the WGD centromere.

The authors state that "The three assemblies and gene models have been submitted to INSDC and are also made available at" - however, the BioProject IDs do not yet have the assemblies associated with this. This might be an oversight in the reporting in the paper (with a separate set of accessions) or the assemblies being processed. The authors should ensure submission of the assemblies and INSDC accession numbers.

Answers to comments from Reviewer #1:

We are grateful to the reviewer for insightful comments and suggestions for improving our manuscript. In particular, we are now able to answer two major concerns. Firstly, we collected two biological replicates from each of the two strains to verify the statistical significance of transcription alternation. Secondly, according to a comment, we showed a large phylogenetic tree of “all” the hyper-/hypo-methylated centromeric repeat regions to reconfirm our finding, with more examples. Let us describe answers to all the comments below.

Comment: The revision failed to address certain major concerns of the study. For example, one major problem raised by this reviewer is that biological replicates for RNA-seq experiment are absolutely required for calling differentially expressed genes and can not be replaced by technical replicates (see comment 1 below).

Answer: We attempted to collect biological replicates. Because the HNI strain is not wild-type but is inbred and weak, it has been difficult for us to have fertile zygotes for multiple biological replicates. We could now have two biological replicates from early embryos (blastulae) for RNA-seq for the HNI and Hd-rR strains. Using these biological replicates, we could have performed subsequent analysis.

Comment. Regarding this reviewer’s comment that the manuscript does not have biological replicates for identifying many (457 out of 716) insertions near the TSS leading to increase of transcription activity. The authors have answered that ‘we did not have biological replicates, we generated technical replicates and used DEseq2 to verify the statistical significance of transcription alternation’. The answer to this comment is not acceptable since technically it is fully feasible to repeat the experiment and carry out RNA-seq with biological replicates for early embryos of Hd-rR and HNI.

Answer: We analyzed RNA-seq data from the two pairs of biological replicates using DESeq2. We detected 101 TSSs that had insertions in their upstream regions and had increased transcript levels in one of the two strains at a stringent statistical significance ($p < 1\%$, Wald test) (Figure 4c,d). These insertions were significantly correlated with increases in the average GC ratio ($p < 10^{-6}$), increases in the average CpG ratios ($p < 10^{-3}$), and decreases in the CpG methylation levels upstream of TSSs ($p=3.9\%$; Extended Data Fig. 10b; Methods).

Comment: In addition, critical information is missing for a newly added analysis Fig. 3D, which led this reviewer to question whether the result was representative.

Comment. Why does Fig. 3D only include a subset of chromosomes?

Comment. The centromeric regions chosen for analysis in Fig. 3D were not justified. What criterion were used for selecting the chromosomes and regions for the analysis? It is unclear how representative these regions are. The length of tested regions ranged from 2.3kb to 166.4kb. What percentage of assembled medaka centromeres were included in this analysis?

Answer: We are sorry that this part was misleading in the previous manuscript. For readability of the phylogenetic tree in Figure 3d, we selected long centromeric regions as representatives. According to the comment, we showed a large phylogenetic tree of “all” centromeric regions in Figure 3d. Although the tree becomes large and complex, it reinforces our finding with more examples.

Comment. The manuscript should explicitly state a conclusion of the analysis shown in Fig. 3D. It seems centromere sequence correlated with chromosomes, and acrocentric vs. non-acrocentric centromeres. What is the conclusion regarding DNA methylation? Is DNA methylation correlated or not correlated with centromere sequence phylogeny?

Answer: DNA methylation patterns in centromeres were not correlated with centromere sequence phylogeny, but hypo/hyper-methylated regions in each chromosome evolved independently. We emphasized this finding in the revision.

Comment. The last section of the study, TSS associated with insertions were examined for CpG methylation and gene expression. However, DNA methylation and gene expression was profiled using different tissues, with the former using germ cells and later using early embryo. The authors need to state this major caveat (tissue mismatch) explicitly in the text, since the decrease in CpG methylation and gene activation may not be correlated when the two features were analyzed using matched tissue.

Answer: Although the description might not be clear, we actually collected both bisulfite data and RNA-seq data from early embryos (blastulae), and observed a significant correlation between the decrease in CpG methylation and the increase in transcripts downstream of TSSs with insertions.

It might be misleading that we also used PacBio reads from testes to call CpG methylation states in highly repetitive regions (e.g., centromeres) where bisulfite sequencing was not helpful. We had to use

testes in place of blastulae to obtain sufficient PacBio reads, because PacBio sequencing demanded DNA from more than one million cells, but we could contain only ~4000 cells per one blastula.

Comment. To “Impact of large-scale insertions...”, Figure 4D and methods - I question the finding that “methylation levels upstream of TSSs with insertions were significantly lower than the levels of those lacking insertions”. In fact, the caption to Figure 4D provides the reason for why I believe this statement is not correct: “methylation ratios are not available for the insert because of high repetitiveness”. Did the analysis pipeline for methylation levels account for this? If no Illumina bisulfite reads can be anchored, what is the methylation result from the PacBio reads that assembled this region/insertion in the first place?

Answer: We are grateful to the reviewer for this insightful comment. We could determine CpG methylation states for ~92% of TSSs with insertions using bisulfite-treated reads from blastulae; however, for the remaining ~8% TSSs, we could not. Figure 4d illustrates such a difficult situation that we could not determine hypo-methylation of a highly repetitive insertion using short bisulfite-treated reads from blastulae, but we could by using PacBio reads from testes. Although testes and blastulae are different tissues, their CpG methylation levels were known to be highly correlated ($R^2=0.73$) in Hd-rR (Qu, W., et al. *Genome Res.*, 22(8), 1419–1425, 2012). These points were not clear in the original manuscripts and were detailed in the revision.

Comment. The manuscript states ‘The centromeres obtained from these germ cells were found to be mostly hypermethylated, which unexpectedly reflects the characteristics of somatic cells.’ However, without presenting any medaka somatic cell methylome in the study, it is unclear how the authors can reach the conclusion that germ cell methylomes are similar to somatic cells.

Answer: To confirm this statement, we used bisulfite-treated reads from testes and liver, germ and somatic cells (Qu, W., et al. *Genome Res.*, 22(8), 1419–1425, 2012). We aligned the reads to the four representative monomers in Extended Data Figure 3b to calculate the average methylation ratios in centromeres in the two tissue types. The average methylation ratio in testes was 72.9%, which was close to 65.3%, the average ratio in liver. We added this confirmation in the main text and methods.

Comment. Figure 3 – A major issue in the first version of this manuscript was the accurate comparison and representation of bisulfite data and SMRT kinetic analysis; While the figures have improved, they are still not convincing because of the low mapping rate of bisulfite reads to the centromeres. The authors should include one panel showing a representative region from a chromosome arm that is highly covered with bisulfite reads and clearly shows congruence to SMRT kinetics.

Answer: We added a figure with three genomic regions in which methylation states determined by AgIn for PacBio sequencing were consistent with those by bisulfite sequencing (Extended Data Figure S4).

Comment 9. Figure 2 is not very meaningful and could be placed entirely into the supplement.

Answer: We replaced Figure 2a with Figure 1d, accordingly. We left Figure 2c in the main figure because it is one of our main findings.

Answers to comments from Reviewer #2:

We would like to thank the reviewer for many insightful comments in the previous manuscript.

Comment: The authors state that "The three assemblies and gene models have been submitted to INSDC and are also made available at" - however, the BioProject IDs do not yet have the assemblies associated with this. This might be an oversight in the reporting in the paper (with a separate set of accessions) or the assemblies being processed. The authors should ensure submission of the assemblies and INSDC accession numbers.

Answer: We put the data into INSDC but we kept them unpublished before the acceptance of the manuscript. We have opened all the deposited data accordingly.

REVIEWERS' COMMENTS:

Reviewer #1 (Remarks to the Author):

The revision has satisfactorily addressed my major concerns. RNA-seq has been generated from biological replicates for early embryos of Hd-rR and HNI. Differentially expressed genes are now analyzed using proper statistical methods. The authors have also improved Fig. 3D to show all chromosomes instead of a subset that was arbitrarily selected.